# SARS-CoV-2 Neutralizing Antibodies Kinetics Postvaccination in Cancer Patients under Treatment with Immune Checkpoint Inhibition

**DOI:** 10.3390/cancers14112796

**Published:** 2022-06-04

**Authors:** Evangelos Terpos, Michalis Liontos, Oraianthi Fiste, Flora Zagouri, Alexandros Briasoulis, Aimilia D. Sklirou, Christos Markellos, Efthymia Skafida, Alkistis Papatheodoridi, Angeliki Andrikopoulou, Konstantinos Koutsoukos, Maria Kaparelou, Vassiliki A. Iconomidou, Ioannis P. Trougakos, Meletios-Athanasios Dimopoulos

**Affiliations:** 1Department of Clinical Therapeutics, National and Kapodistrian University of Athens, Alexandra Hospital, 11528 Athens, Greece; mlionto@med.uoa.gr (M.L.); ofiste@med.uoa.gr (O.F.); fzagouri@med.uoa.gr (F.Z.); abriasoulis@med.uoa.gr (A.B.); krisnm@med.uoa.gr (C.M.); efiskafida@med.uoa.gr (E.S.); alkipapath@med.uoa.gr (A.P.); aggandrikop@med.uoa.gr (A.A.); konkoutsoukos@med.uoa.gr (K.K.); mkaparelou@yahoo.com (M.K.); mdimop@med.uoa.gr (M.-A.D.); 2Section of Cell Biology and Biophysics, Department of Biology, School of Sciences, National and Kapodistrian University of Athens, Panepistimiopolis, 15701 Athens, Greece; asklirou@biol.uoa.gr (A.D.S.); veconom@biol.uoa.gr (V.A.I.); itrougakos@biol.uoa.gr (I.P.T.)

**Keywords:** SARS-CoV-2, vaccination, cancer, immune checkpoint inhibitors, immunotherapy

## Abstract

**Simple Summary:**

Solid tumor patients under active anticancer treatment are peculiarly affected by COVID-19 infection, given not only its ominous outcomes but also the need of disruptions of their rather strict therapeutic scheme. Thus, they have been globally prioritized for both primary and booster vaccinations. The existing data with respect to the seroconversion rate of neutralizing antibodies (NAbs) among them, after vaccination, remain nevertheless obscure. Therefore, we prospectively evaluated the long-term humoral immunity dynamics for up to one month after the third dose in patients with solid malignancies receiving immunotherapy. Further research is required to assess the incremental benefit of booster doses and to optimize the vaccination schedule across different types of cancer and diverse systemic therapies.

**Abstract:**

Considering that COVID-19 could adversely affect cancer patients, several countries have prioritized this highly susceptible population for vaccination. Thus, rapidly generating evidence on the efficacy of SARS-CoV-2 vaccination in the subset of patients with cancer under active therapy is of paramount importance. From this perspective, we launched the present prospective observational study to comprehensively address the longitudinal dynamics of immunogenicity of both messenger RNA (mRNA) and viral vector-based vaccines in 85 patients treated with immune checkpoint inhibitors (ICIs) for a broad range of solid tumors. Despite the relatively poor humoral responses following the priming vaccine inoculum, the seroconversion rates significantly increased after the second dose. Waning vaccine-based immunity was observed over the following six months, yet the administration of a third booster dose remarkably optimized antibody responses. Larger cohort studies providing real-world data with regard to vaccines effectiveness and durability of their protection among cancer patients receiving immunotherapy are an increasing priority.

## 1. Introduction

Since March 2020, when the World Health Organization (WHO) declared the novel SARS-CoV-2-associated coronavirus disease 2019 (COVID-19) a global pandemic [1], more than 512 million confirmed cases and 6.23 million deaths have been reported worldwide [2]. This led to an unprecedented scientific effort that resulted in the prompt characterization of the viral genome [3] and structure [4], as well as its interaction with host cells [5]. During the pandemic, it allowed for a rapid development of both prophylactic vaccines, in less than a year, and therapeutic agents that are currently used in clinical practice. Up to March 2022, five vaccines had shown clinical efficacy and were approved by the European Medical Agency (EMA), namely BNT162b2, developed by Pfizer BioNTech [6]; NVX-CoV2373, developed by Novavax [7]; mRNA-1273, developed by Moderna; NIAID [8], AZD1222, developed by the University of Oxford and AstraZeneca [9]; and AD26.COV2-S, developed by Janssen Pharmaceutical Companies, offering a glimpse of hope for a return to normality [10]. The development of safe and effective vaccines displays a pivotal step towards preventing not only pandemic exacerbation, but also severe COVID-19 infection in the immunocompromised, including oncological patients, who have therefore been prioritized for vaccination [11].

Indeed, from early on during this pandemic, cancer patients have been identified as vulnerable subjects prone to both severe disease and death, presumably as a result of their impaired immune system, by the underlying disease itself and/or the required myelosuppressive anticancer treatments [12,13]. Therefore, guidelines were issued for the optimal management of cancer patients during the pandemic, both at an international [14] and local level [15]. In addition, cancer patients were strongly advised to undergo full-schedule vaccination for COVID-19 [16]. However, current knowledge with respect to the safety, tolerability, and efficacy of the EMA-authorized vaccines in patients with cancer under active treatment is limited, as these subjects were not enrolled in the confirmatory trials. Indeed, the required time to develop immunity and its duration, alongside the impact of distinct anticancer regimens on this immunity, as well as the optimal vaccination schedule, all remain uncertain within the oncological community and will probably be promptly addressed in post-license, real-world studies.

Considering that neutralizing antibody (NAbs) levels have been correlated with clinically relevant immune protection against SARS-CoV-2 variants [17], we undertook a prospective study (NCT04743388) in order to investigate the antibody responses after COVID-19 vaccination in patients with solid tumors, hematological malignancies, and healthy volunteers [18]. We have already published the results of early immunological responses post first dose vaccination in cancer patients treated with immune checkpoint inhibitors (ICIs) who exhibited a blunted humoral response compared with matched healthy volunteers [19]. Herein, we prospectively evaluated the kinetics of NAbs directed against the SARS-CoV-2 spike-receptor binding domain for up to one month after the administration of a booster vaccine dose in this cohort of patients under immunotherapy for multiple solid malignancies.

## 2. Materials and Methods

### 2.1. Patients

Enrollment criteria for the monocentric NCT04743388 study included healthy volunteers and all individuals who, according to the instructions of the Greek government, are considered eligible to receive vaccination for COVID-19. Major inclusion criteria for the patient cohort of this study included: (i) age above 18 years; (ii) the presence of histologically and/or cytologically confirmed solid malignant neoplasm treated with ICIs as per standard of care; (iii) the capacity to sign an informed consent form; and (iv) eligibility for SARS-CoV-2 vaccination. Volunteers matched for age (1:1) with no active malignant disease were used as a control group. Major exclusion criteria for both patient- and control cohorts included the presence of: (i) other active malignant disease; (ii) autoimmune disease; (iii) Human Immunodeficiency Virus (HIV) and/or active hepatitis B and C infection; and (iv) prior diagnosis of COVID-19 infection using a polymerase chain reaction (PCR) test.

The study was approved by the respective Ethical Committees in accordance with the Declaration of Helsinki and the International Conference on Harmonization for Good Clinical Practice. All patients and controls provided written informed consent prior to enrollment in the study. The confidentiality of the participants’ data was maintained in accordance with the rules of the General Data Protection Regulation (GDPR). All of the subject’s identities were kept strictly private following the principles of ‘pseudonymisation’.

### 2.2. Neutralizing Antibodies Detection

NAbs against ancestral SARS-CoV-2 variants (Wuhan-Hu-1) had been prospectively determined, using a U.S. Food and Drug Administration (FDA)-approved enzyme-linked immunosorbent assay (ELISA; cPass™ SARS-CoV-2 NAb Detection Kit; GenScript, Piscataway, NJ, USA), in serial plasma samples, after vein puncture, at several timepoints; at baseline (D1; before the first dose), prior to second vaccination (D22), one month post second dose (D50), three months post second dose (D90), six months post second dose (D180), and one month after the administration of a booster vaccine dose (3rd + 30D). Serum was separated within four hours of blood collection and stored at −80 °C until the day of measurement. Seropositivity was considered as a result of ≥30%, while a Nab titer of at least 50% was associated with clinically relevant viral inhibition, as previously suggested [20,21]. Samples from the same patient or control were measured on the same ELISA plate.

### 2.3. Outcomes

The primary endpoint of this study was to examine the longitudinal NAb responses following immunization for up to one month post booster dose.

### 2.4. Statistical Analysis

Baseline demographics, co-morbidities, and the Nab levels were compared between the two groups, and a chi-square test for categorical variables and an unpaired *t*-test or Wilcoxon signed-rank test were used (as appropriate) for continuous variables. Mixed models were performed using direct likelihood estimation with fixed effects of antibody titers, timing of measurement, and interaction of antibody titers by timing of measurement. An unstructured covariance matrix was used to model within-patient error. All data extraction and statistical analyses were conducted using Stata Version 17.0 (StataCorp LLC., College Station, TX, USA). Case control matching to match the two groups for age and BMI was used with the calipmatch command in Stata. All significance tests were two- tailed and conducted at the 5% significance level.

## 3. Results

Between 28 January 2021 and 4 February 2022, a total of 160 participants enrolled in the present analysis; 85 patients with cancer and a median age of 68.04 years (IQR: 62–77 years) and 75 controls (median age 65.51 years, IQR: 62–68 years; *p* = 0.12 for age compared with patients). In the patient group, the majority were of male gender (52; 61.18%), while among the controls, 44 (58.67%) were female (*p* = 0.012 for gender compared with patients). The median body mass index (BMI) was 26.89 kg/m^2^ and 26.88 kg/m^2^ for cancer patients and for controls, respectively (*p* = 0.47). Among patients, genitourinary was the most common cancer type (50.59%), followed by lung (23.53%), and gynecological/breast (14.12%).

At the time of first vaccination, 66 patients (77.65%) were on active treatment with anti-programmed death-1 (PD-1) antibodies, whereas 11 (12.94%) were on anti-programmed death-ligand-1 (PD-L1), and eight (9.41%) received immuno-oncology (I/O) combos. More specifically, 41 patients were under Pembrolizumab (anti-PD-1), 25 were under Nivolumab (anti-PD-1), seven were under Atezolizumab (anti-PD-L1), two under Avelumab (anti-PD-L1), two were under Durvalumab (anti-PD-L1), and eight were receiving Nivolumab + Ipilimumab (anti-PD-1 and antibody against cytotoxic lymphocyte-associated protein 4; anti-CTLA4, combo). Moreover, the vast majority of the patients were on first (52/85; 61.18%) and second (18/85; 21.18%) lines of treatment. The median time between vaccine administration and active treatment with immunotherapy was seven days (range 5–10 days) prior to therapy. Comorbidities in the patients’ cohort included cardiovascular disease (49.41%), diabetes mellitus (20%), and pulmonary disease (10.59%). Sixty-one (71.76%) patients had been vaccinated with the BNT162b2 vaccine, whereas 17 (20%) subjects received the AZD1222 vaccine. At the time of booster dose vaccination, 68 (80%) patients were alive, of whom 52 (76.47%) were on active I/O treatment. Summary of the main characteristics of the 85 enrolled patients are depicted in Table 1, whereas a case-control comparison with regard to baseline characteristics are provided in Appendix A.

At baseline (D1), the NAb titers did not differ between the two groups (median 15.82% for cancer patients versus 14.34% for healthy controls; *p* = 0.14). On D22 after the first dose, NAb titers significantly increased in both cohorts (*p* = 0.016); however, the mean NAbs were 29.08% in cancer patients and 44.15% in the control group (*p* < 0.001). Indeed, only 28.2% of patients, compared with 49.3% of controls (*p* = 0.006), developed clinically relevant viral inhibition after the initial vaccine shot. On D50 (one month post second dose vaccination) a further increase in NAbs was evident for both patients and controls (70.73% and 91.74%, respectively; *p* < 0.001). Thus, after full vaccination, 73.5% of patients, and 98.6% of healthy volunteers, developed NAb titers ≥50% (*p* < 0.001).

From this timepoint on, a gradual but steady decline was observed, with the mean NAb titers being 61.27% for patients and 84.65% for controls at D90 (*p* < 0.001), and 48.45% versus 72.16% for each group, respectively, at D180 (*p* = 0.001). Crucially, such findings could imply attenuated protection against COVID-19 in fully vaccinated patients receiving active anticancer treatment. Nevertheless, one month after the administration of a booster dose (third + 30D) the humoral response significantly improved. More specifically, the antibody response reached 96.6% in the patients’ subgroup (median NAbs, 97.5%; IQR: 96.5–97.9), in comparison with 96.3% for controls (*p* = 0.68). Table 2 displays both the mean and median NAb titers in cancer patients under ICIs at the aforementioned timepoints, while Figure 1 demonstrates the percent inhibition of NAbs across the timeline of vaccination, from baseline to one month after the boosted immunization.

Regarding the immunological response, there was no statistically significant difference in the Nab levels according to type of immunotherapy received. No statistically significant difference was also noted at any time point for all variables analyzed, including age, BMI (≤25 vs. >25), sex, type of cancer, type of vaccine, and comorbidities. Additionally, our cohort included three patients who had been infected with COVID-19 prior to second dose, two of whom had been vaccinated with AZD1222 and who did not seroconvert one month post the completion of the immunization scheme. On the contrary, the third patient who received the mRNA BNT162b2 vaccine had a high rate of seroconversion after the second dose.

The vaccines were in general well tolerated among all participants, in opposition to the anticipated exaggerated immune responses. Common adverse events in cancer patients included pain at the injection site (31.76%), fatigue (20%), and fever (8.24%).

## 4. Discussion

The emergence of COVID-19 since December 2019 has imposed heavy costs on healthcare systems, economies, and societies worldwide [22]. Apart from the basic preventive measures of social distancing and mask wearing, vaccination represents a rather significant milestone in global efforts to mitigate the impacts of this pandemic. As of 27 April 2022, 4.64 billion individuals had been fully vaccinated worldwide [23]. Yet, to the best of our knowledge, the long-term efficacy of the mRNA and viral vector COVID-19 vaccines in the fragile population of actively treated with immunotherapy cancer patients has not been explicitly delineated.

Since March 2011, when ipilimumab, a monoclonal antibody (mAb) directed against cytotoxic lymphocyte-associated protein 4 (CTLA-4), became the first FDA-approved ICI for patients with advanced/metastatic melanoma, numerous mAbs targeting other checkpoints (like PD-1 and PD-L1) have been implemented as another pillar in cancer clinical practice, providing durable responses in a wide array of solid malignancies. Given the ICIs’ pleiotropic effects on immunity, several key questions regarding their interaction with SARS-CoV-2 remain to be promptly answered. In particular, the enhanced immunomodulation, driven by immunotherapy, could contribute to higher levels of viral clearance, yet it could also lead to hyperinflammatory responses and thus worse clinical outcomes from COVID-19 infection [24].

In this respect, acknowledging that cancer patients receiving immunotherapy are in double jeopardy due to the high risk of severe infection and the dubious interaction between treatment with ICIs and COVID-19 vaccination, we had previously evaluated both the safety and efficacy of the first vaccination dose in 59 patients under immune checkpoint inhibition [19]. Consequently, the present single-center, cohort study aimed to prospectively examine the kinetics of NAbs against SARS-CoV-2 after full vaccination for up to one month post third dose, in 85 patients with solid tumors, who were receiving ICIs. Furthermore, a recent retrospective study of immunogenicity and safety of mRNA COVID-19 vaccine among 326 actively treated solid tumor patients highlighted the decreased seroconversion rates among those under chemotherapy compared with those receiving targeted therapies or immunotherapy [25].

Our report suggests not only high rates of gradual decline in NAbs six months after the second dose of COVID-19 vaccines, but also the optimization of humoral immunogenicity against SARS-CoV-2 after the administration of a booster dose, in cancer patients under immune checkpoint inhibition. Indeed, after the third dose no patients and controls were seronegative. These findings are consistent with previously published research on the antibody kinetics in cancer patients after a period of a few months following full vaccination [25,26,27,28,29,30]. Furthermore, a recently published meta-analysis demonstrated reduced seroconversion rates after one COVID-19 vaccine dose in patients with solid cancers (Risk Ratio: 0.55; 95% confidence interval: 0.46–0.65; I2 = 78%; Absolute Risk: 0.44; 95% CI: 0.36–0.53; I2 = 84%) in comparison with immunocompetent controls; yet, the administration of a second dose led to significantly increased seroconversion (Risk Ratio: 0.90; 95% CI: 0.88–0.93; I2 = 51%; Absolute Risk: 0.89; 95% CI: 0.86–0.91; I2 = 49%), whereas the 3rd dose was associated with enhanced humoral response in vaccine non-responders with solid tumors, further highlighting the benefits of a booster dose [31].

Moreover, taking into consideration the emerging SARS-CoV-2 variants of concern, primary immunization with BNT162b2 or AZD1222 provided insufficient protection against mild and/or symptomatic infection with the omicron (or B.1.1.529) variant, whereas a booster dose with either BNT162b2 or mRNA-1273 resulted in a significantly increased, yet waning over time, immune response [32]. Lauring Adam and colleagues consistently reported that two primary doses of mRNA vaccination were less effective against COVID-19 hospitalization related to the omicron variants compared with infections caused by alpha (or B.1.1.7) and delta (or B.1.617.2) variants [33]. A recent retrospective study from Qatar similarly highlighted the need of a third mRNA booster dose, as it led to meaningful protection against hospitalization and death by either subvariant (delta and omicron) [33]. The authors also emphasized the necessity of development of next-generation vaccines which could not only be able to target a broad range of SARS-CoV-2 subtypes, but also protect from novel variants, preempting the next global pandemic [34].

Furthermore, the decision on a second booster dose remains at present controversial. Three Israeli studies have examined the immunogenicity of a fourth mRNA booster dose; the first which enrolled 274 healthy healthcare workers reported a rather marginal benefit, as it offered a partial defense against the omicron variants [35], while the other two which included adults 60 years of age or older demonstrated a substantial reduction of confirmed SARS-CoV-2 infection cases (including severe illness) and hospitalizations or deaths due to COVID-19, respectively [36,37]. Apart from the lack of mature data with regard to the effectiveness of multiple booster shots, the EMA’s head of vaccines strategy, Mr. Marco Cavaleri, had recently raised concerns of T-cell exhaustion, thus weakening the immune responses, with frequent (i.e., every four months), additional doses [38]. Currently, the administration of a fourth shot follows a rather precautionary principle, while academic institutions, non-profit organizations, government agencies, and biotechnology companies are in pursuit of a safe and effective ‘pan-coronavirus’ vaccine with increased breadth of coverage.

Several shortcomings of the present study need to be listed. These include the lack of data regarding SARS-CoV-2-specific T-cell mediated immune responses, which could be of utmost importance for protection from COVID-19. Apart from the underestimation of cellular immunity, the heterogeneous (regarding underlying tumor- and ICIs- types) sample size is relatively limited to draw any strong conclusion with regards to seroconversion rates. Moreover, the inclusion of subjects from one only hospital does not permit representativeness of the patient population, and thus a selection bias is possible. Furthermore, the two cohorts of patients and volunteers were not optimally matched with regards to both age, sex, and type of COVID-19 vaccine, thereby posing the risk of confounding. It is worthy of note that it has been suggested that male gender relates to lower humoral response to mRNA vaccines [39]. Nevertheless, we firmly believe that these results represent a real-world scenario advocating for a booster shot in patients with solid neoplastic diseases under systemic treatment, which along with the implementation of protective measures among patients and healthcare providers [40] could ensure continuity of cancer care during the pandemic [41]. Further validation in larger cohorts is warranted to further narrow the knowledge gap, as it could fully elucidate the efficacy and safety of these vaccines and prudently inform public health policies with respect to the ideal timing of their administration in this complex population.

## 5. Conclusions

The present prospective observational study aimed to longitudinally assess the serum Nabs titers of 85 cancer patients under active treatment with immunotherapy up to one-month post-vaccination with a booster shot. Despite the rather delayed mobilization of SARS-CoV-2 humoral responses after the initial vaccination, solid tumor patients receiving ICIs (compared with healthy individuals) develop robust immunogenicity after the second dose. Importantly, the waning durability of protection, following a full primary vaccination course, underscores the need of an additional booster shot to sustain protection against moderate and severe COVID-19 infection, especially in this high-risk population.

## Figures and Tables

**Figure 1 cancers-14-02796-f001:**
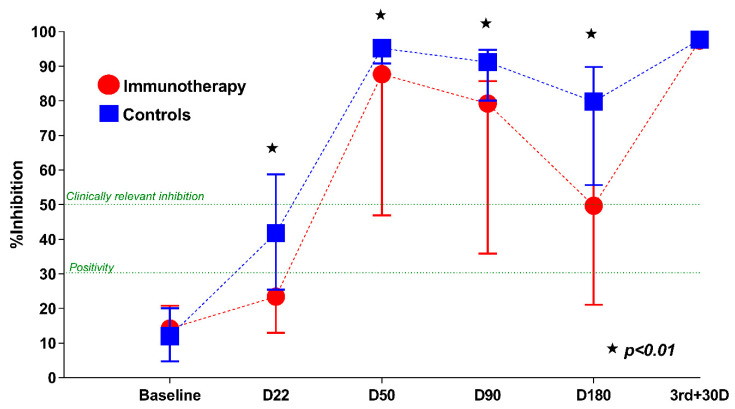
Kinetics of neutralizing antibodies (NAbs) in cancer patients receiving immunotherapy and matched controls at several timepoints after COVID-19 vaccination. The asterisk (★) denotes a statistically significant difference between the sixth month levels and the previous timepoints (*p* < 0.01).

**Table 1 cancers-14-02796-t001:** Baseline patient characteristics.

Variables	Total Population; Median (IQR)
Age	68.04 (62–77)
BMI	26.89 (24–29)
**Sex**	
Male	52 (61.18%)
Female	33 (38.82%)
**Type of Cancer**	
Urothelial/bladder cancer	22 (25.89%)
Renal cancer	21 (24.71%)
Lung cancer	20 (23.53%)
Endometrial cancer	5 (5.88%)
Pancreatic cancer	2 (2.35%)
Other	15 (17.65%)
**Type of Therapy**	
Anti-PD-1	66 (77.65%)
Anti-PD-L1	11 (12.94%)
I/O combo	8 (9.41%)
**Comorbidities**	
Yes	64 (75.29%)
None	13 (15.29%)
Missing	8 (9.41%)
**Type of Vaccine**	
BNT162b2	61 (71.76%)
AZD1222	17 (20%)
mRNA-1273	7 (8.24%)
**COVID-19 Infection**	
Yes	3 (3.53%)
No	78 (91.76%)
Missing	4 (4.71%)
**Vaccine-Related Adverse Events**	
Pain at injection site	27 (31.76%)
Fatigue	17 (20%)
Fever	7 (8.24%)
None	36 (42.35%)
Other or missing	14 (16.47%)

**Table 2 cancers-14-02796-t002:** Neutralizing antibody (NAb) titers in cancer patients receiving immunotherapy at several timepoints after COVID-19 vaccination.

Timepoint	Mean NAb Titers ± Standard Deviation	Median NAb Titers (IQR)
Baseline (D1)	15.8 ± 9.4	14.2 (10.7–20.8)
Prior to 2nd dose (D22)	29.1 ± 24.7	23.4 (13–39.8)
1 month after 2nd dose (D50)	70.7 ± 31.8	87.1 (46.9–95.9)
3 months after 2nd dose (D90)	61.3 ± 28.9	70.2 (35.9–85.7)
6 months after 2nd dose (D180)	48.4 ± 28.2	49.7 (21.1–78.8)
1 month after 3rd dose (3rd + 30D)	96.6 ± 2.1	97.5 (96.5–97.9)

IQR: interquartile range.

## Data Availability

All data generated or analysed during this study are included in this published article.

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
