# Peer review of "SARS-CoV-2 Neutralizing Antibodies Kinetics Postvaccination in Cancer Patients under Treatment with Immune Checkpoint Inhibition"

_cancers, 2022, doi:10.3390/cancers14112796_

Round 1
Reviewer 1 Report
The presented study is interesting and shows the benefit of vaccination in patients undergoing immunotherapy. I would like to ask for comparison and description of specific drugs and taking into account the lines of treatment. Is there any particular patient population that responds more or less to vaccination?
Author Response
We would like to thank the reviewer for his/her positive comments in our study.
In the revised version of the manuscript we have added treatments of patients as well as line of therapy they were received. We added the sentence ‘’More specifically, 41 patients were under Pembrolizumab (anti-PD-1), 25 under Nivolumab (anti-PD-1), 7 under Atezolizumab (anti-PD-L1), 2 under Avelumab (an-ti-PD-L1), 2 under Durvalumab (anti-PD-L1), whereas 8 were receiving Nivolumab + Ipilimumab (anti-PD-1 and antibody against cytotoxic lymphocyte-associated protein 4; anti-CTLA4, combo). Moreover, the vast majority of the patients were on first (52/85; 61.18%) and second (18/85; 21.18%) line of treatment.’’ (2nd paragraph of Results).
We also used the following phrase ‘’Regarding the immunological response, there was no statistically significant difference in the Nab levels according to type of immunotherapy received.’’ (5th paragraph of Results).
Reviewer 2 Report
The study is interesting in that it shows that treatment with immunotherapy does not decrease the efficacy of boost compared to a control population. The letter is well written. Some additional data are missing to better interpret these results
- The vaccine was administered at what time on average after the immunotherapy treatment
- Does the response to boost vary between the different immunotherapy treatments (anti-PD-1/anti-PD-L1/Anti-CTLA-4). Please also indicate the respective frequency of these treatments at the time of the boost
- It is indicated that the Nab assay is performed against the variants. Would it be possible to specify against which variants the Nab was detected and if it is possible to extend the results of figure 1 to the detection of other variants including the Omicron strain.
Author Response
We would like to thank the reviewer for his/her positive comments!
- The vaccine was administered at what time on average after the immunotherapy treatment
The sentence ‘’The median time between vaccine administration and active treatment with immunotherapy was 7 days (range 5-10 days), prior therapy.’’ was added in the 2nd paragraph of Results.
- Does the response to boost vary between the different immunotherapy treatments (anti-PD-1/anti-PD-L1/Anti-CTLA-4). Please also indicate the respective frequency of these treatments at the time of the boost.
We thank the reviewer for this thoughtful comment. As previously described there was no statistical significant difference in the immunological response among immunotherapy treatments. Specifically, the statistical significance of this analysis is limited regarding the booster dose due to the small number of patients that performed booster dose. However, it is already stated in our manuscript that the antibody response reached 96.6% in the patients’ subgroup (median NAbs, 97.5%; IQR: 96.5-97.9), in comparison with 96.3% for controls (p=0.68) after the booster dose. These results are regardless of ICIs used.
- It is indicated that the Nab assay is performed against the variants. Would it be possible to specify against which variants the Nab was detected and if it is possible to extend the results of figure 1 to the detection of other variants including the Omicron strain.
This is a very important issue raised by the reviewer. Our analysis was performed again the initial Wuhan variant and this is explained in details in the revised version of the manuscript. More specifically, ‘’NAbs against ancestral SARS-CoV-2 variants (Wuhan-Hu-1) had been prospectively determined’’ was added in paragraph 2.2.
Reviewer 3 Report
In the manuscript entitled "SARS-CoV-2 neutralizing antibodies kinetics postvaccination in cancer patients under treatment with immune checkpoint inhibition," Evangelos Terpos and colleagues set out to study the kinetics of SARS-CoV-2 neutralizing antibodies (NAbs) in cancer patients undergoing immunotherapy treatment.
A total of 160 patients were enrolled in this prospective study, comparing 85 cancer patients with 75 healthy volunteers in the control group. The authors had the purpose to measure antibody response levels in the group of cancer patients undergoing immunotherapy treatment and in controls up to one month after administration of the third (booster) dose of vaccine.
The paper is well written and the results are clearly presented. Graphic is intuitive and immediately understandable. Moreover, in agreement with what the authors reported in the discussion, the paper is particularly interesting because it explores the level of antibody response in cancer patients even after the third dose, while most papers yet published in the literature stop at the first two doses.
As a major limitation, the study provides results of a relatively small, monocentric cohort of cancer patients treated with immune checkpoint inhibitors, thus limiting the robustness of evidence provided. While this is unlikely to be corrected, I would suggest some major corrections that could improve the level of evidence:
- First, the monocentric nature of the investigation should be clearly declared in the methods
- In the Introduction, the authors do not explain the rationale to include only patients treated with ICI. Is there any? Is there an available cohort treated with standard chemotherapy to compare with?
- Were all patients treated with ICI as per standard of care, or within clinical trials?
- The authors declare that healthy controls were matched based on age and sex. However, the % of male/females is significantly different. How so? In literature, there are reviews indicating that male sex is related to a lower humoral response to mRNA vaccines. (Notarte KI, Ver AT, Velasco JV, Pastrana A, Catahay JA, Salvagno GL, Yap EPH, Martinez-Sobrido L, B Torrelles J, Lippi G, Henry BM. Effects of age, sex, serostatus, and underlying comorbidities on humoral response post-SARS-CoV-2 Pfizer-BioNTech mRNA vaccination: a systematic review. Crit Rev Clin Lab Sci. 2022 Feb 28:1-18. doi: 10.1080/10408363.2022.2038539. Epub ahead of print. PMID: 35220860; PMCID: PMC8935447.). Also, differences in terms of ICI efficacy are reported (Cancer immunotherapy efficacy and patients’ sex: a systematic review and meta-analysis. Fabio Conforti et al. Lancet Oncol. 2018 Jun; 19(6):737-746.). The authors should discuss this.How was the matching performed? Did they adopt a propensity score matching method? Also, I would suggest to match cases and controls also by the type of vaccine, as differences in the literature have been reported.
- Similarly, I kindly ask to provide a descriptive table comparing cases and controls (with the exception of type of cancer and anticancer treatment, obviously)
- As an additional study endpoint, could the authors provide data regarding the rate of patients and controls that developed COVID19? This is the major role of vaccines and a valuable surrogate endpoint.
- It would be interesting,as secondary endpoint, to investigate whether the variables you considered (Age, BMI, Sex, Type of cancer, type of therapy, comorbidities, type of vaccine) are related to differences in seroconversion or antibody titer value. I would also add the severity of the patients, assessed with the ECOG scale.
- comparison with other european experiences in the discussion is suggested (I would suggesthttps://pubmed.ncbi.nlm.nih.gov/33158968/ or https://pubmed.ncbi.nlm.nih.gov/32659475/)
Based on the above considerations, I recommend major revision before publication of the manuscript.
Author Response
Thank you for the insightful and particularly valuable comments.
- First, the monocentric nature of the investigation should be clearly declared in the methods
In accordance with reviewer’s suggestion, we have added in the revised version of the manuscript the following sentence in the Methods section: ‘’Enrollment criteria for the monocentric NCT04743388 study’’ (paragraph 2.1.). Moreover, we include this aspect among the shortcomings of our study (7th paragraph of Discussion).
- In the Introduction, the authors do not explain the rationale to include only patients treated with ICI. Is there any? Is there an available cohort treated with standard chemotherapy to compare with?
It is known that chemotherapy compromises immune response and reduces responses to vaccination. This has also been proved for COVID-19 vaccination and we have included in the revised version of the manuscript a related text: ‘’Besides, a recent retrospective study of immunogenicity and safety of mRNA COVID-19 vaccine among 326 actively treated solid tumor patients highlighted the decreased seroconversion rates among those under chemotherapy, compared with those receiving targeted therapies or immunotherapy [25].’’ (3rd paragraph of Discussion).
Targeted therapies and immune checkpoint inhibitors are commonly used in several neoplasms and probably will not affect immune response to vaccination. Thus, our study focused on these agents. We have stated at the 3rd paragraph of Discussion that ‘’acknowledging that cancer patients receiving immunotherapy are in double jeopardy, due to the high risk of severe infection and the dubious interaction between treatment with ICIs and COVID-19 vaccination, we had previously evaluated both the safety and efficacy of the first vaccination dose in 59 patients under immune checkpoint inhibition [19]. Consequently, the present single-center, cohort study aimed to prospectively ex-amine the kinetics of NAbs against SARS-CoV-2 after full vaccination for up to 1 month post 3rd dose, in 85 patients with solid tumors, who were receiving ICIs.’’
At the present study we did not include any cohort of cancer patients under chemotherapy since this was out of the scope of our study.
- Were all patients treated with ICI as per standard of care, or within clinical trials?
All patients were treated as per standard of care and this is now stated in the revised version of the manuscript. We added “as per standard of care” in the inclusion criteria for the patient cohort selection (paragraph 2.1.)
- The authors declare that healthy controls were matched based on age and sex. However, the % of male/females is significantly different. How so? In literature, there are reviews indicating that male sex is related to a lower humoral response to mRNA vaccines. (Notarte KI, Ver AT, Velasco JV, Pastrana A, Catahay JA, Salvagno GL, Yap EPH, Martinez-Sobrido L, B Torrelles J, Lippi G, Henry BM. Effects of age, sex, serostatus, and underlying comorbidities on humoral response post-SARS-CoV-2 Pfizer-BioNTech mRNA vaccination: a systematic review. Crit Rev Clin Lab Sci. 2022 Feb 28:1-18. doi: 10.1080/10408363.2022.2038539. Epub ahead of print. PMID: 35220860; PMCID: PMC8935447.) Also, differences in terms of ICI efficacy are reported (Cancer immunotherapy efficacy and patients’ sex: a systematic review and meta-analysis. Fabio Conforti et al. Lancet Oncol. 2018 Jun; 19(6):737-746.). The authors should discuss this. How was the matching performed? Did they adopt a propensity score matching method? Also, I would suggest to match cases and controls also by the type of vaccine, as differences in the literature have been reported.
We would like to thank the reviewer for his comment. Indeed, matching has been performed with age but there are differences in the percentage of male/females among the two cohorts. Thus, we have replaced the terminology in Materials and Methods section to ‘’matched for age” instead of “matched for age and gender” (paragraph 2.1., Materials and Methods). We have also commented on the effect of sex on the humoral response to mRNA vaccines as suggested by the reviewer: ‘’Furthermore, the two cohorts of patients and volunteers were not optimally matched, with regards to both age, sex, and type of COVID-19 vaccine, posing the risk of confounding. Noteworthy, it has been suggested that male gender relates to lower humoral response to mRNA vaccines [39].’’ (7th paragraph of Discussion). Case control matching to match the two groups for age and BMI was used with the calipmatch command in Stata. This is now added in the revised version of the manuscript.
- Similarly, I kindly ask to provide a descriptive table comparing cases and controls (with the exception of type of cancer and anticancer treatment, obviously)
We thank the reviewer for this proposal. However, we believe that all these data are described already in the manuscript and such a Table would not add to the already presented data.
- As an additional study endpoint, could the authors provide data regarding the rate of patients and controls that developed COVID19? This is the major role of vaccines and a valuable surrogate endpoint.
This is a very interesting issue raised by the reviewer. In our cohort, three patients were diagnosed positive for COVID-19. Two of them were vaccinated with AZD1222 and one with BNT162b2 mRNA vaccine. This information is now added in the results section of the revised version of the manuscript. More specifically, the sentences ‘’Additionally, our cohort included 3 patients who had been infected with COVID-19 prior to second dose, two of whom had been vaccinated with AZD1222 and who did not seroconvert one month post the completion of the immunization scheme. On the contrary, the third patient who received the mRNA BNT162b2 vaccine had a high rate of seroconversion after the second dose.’’ were added in the 5th paragraph of Results.
- It would be interesting, as secondary endpoint, to investigate whether the variables you considered (Age, BMI, Sex, Type of cancer, type of therapy, comorbidities, type of vaccine) are related to differences in seroconversion or antibody titer value. I would also add the severity of the patients, assessed with the ECOG scale.
As suggested by the reviewer we have performed this analysis for the revised version of the manuscript. No statistically significant difference was noted at any time point for all variables analyzed (Age, BMI (≤25 vs >25), Sex, Type of cancer, type of therapy and type of vaccine and comorbidities). No data were available for baseline ECOG performance status. These data are now included in the results section of the revised version of the manuscript.
- comparison with other european experiences in the discussion is suggested (I would suggest https://pubmed.ncbi.nlm.nih.gov/33158968/ or https://pubmed.ncbi.nlm.nih.gov/32659475/)
We thank the reviewer for his valuable input. We have included the suggested references in the revised version of the manuscript.
Once again, we thank the reviewers for their careful reading of our manuscript and their constructive remarks!
Round 2
Reviewer 2 Report
The authors have satisfactorily address my various concerns
Author Response
We are grateful for your consideration of this manuscript!
Reviewer 3 Report
I thank the authors for improving their manuscript with further data, I appreciate the effort.
I would still suggest to add a descriptive table comparing cases and controls.
With that, the manuscript is suitable for publication.
Author Response
We appreciate the reviewer's positive feedback and as suggested, we have also included Supplementary Table 1. with case-control comparison regarding main baseline characteristics (age, BMI, and gender).